# Viewing Nature Lets Your Mind Run Free: Three Experiments about the Influence of Viewing a Nature Video on Cognitive Coping with Psychological Distress

**DOI:** 10.3390/ijerph18168842

**Published:** 2021-08-22

**Authors:** Daphne Meuwese, Jolanda Maas, Lydia Krabbendam, Karin Dijkstra

**Affiliations:** 1Department of Clinical, Neuro and Developmental Psychology, Vrije Universiteit Amsterdam, 1081 BT Amsterdam, The Netherlands; j2.maas@vu.nl (J.M.); lydia.krabbendam@vu.nl (L.K.); 2Research Group Nursing, Saxion University of Applied Sciences, 7513 AB Enschede, The Netherlands; k.dijkstra@saxion.nl

**Keywords:** restorative environments, coping, clinical psychology, state coping scale (SCS), mental health

## Abstract

Viewing nature has restorative qualities that might help people cope with their personal struggles. Three lab experiments (*N* = 506) studied whether environment (nature vs. built) influences cognitive coping with psychological distress. Psychological distress was induced with an autobiographical recall task about serious regret, whereafter participants were randomly assigned to view a nature or built video. Cognitive coping (i) Quantity, (ii) Content, and (iii) Quality were hereafter assessed as well as extent and vividness of the regretful memory during the video. Results showed a higher cognitive coping Quantity (Study 1 and 3) and a higher cognitive coping Quality (All studies) for the nature (vs. built) condition. Regarding cognitive coping Content, results varied across the studies. Additionally, participants reported to have thought about the experienced psychological distress to a greater extent while viewing the nature (vs. built) video. Yet they did rate viewing nature as more relaxing. We propose a two-step pathway as an underlying mechanism of restoration. In the first step the capacity for directed attention replenishes. Secondly, this renewed capacity is directed towards internal processes, creating the optimal setting for reflection. Hence, viewing nature allows people to truly process whatever is occupying their minds, which is ultimately relieving and beneficial for mental health.

## 1. Introduction

“*And into the forest I go to lose my mind and find my soul*”. John Muir

When weighed down by personal struggles, we might recognize the longing for something to provide a sense of relief and help us cope with the experienced psychological distress. Natural environments might provide such a safe haven where we can catch our breath and process whatever is occupying our minds. It is well-established that the natural environment has restorative effects such as recovery from stress, [1], mood enhancement [2], and replenishment of attention [3], that benefit mental health. To exemplify, when asked about their experiences with nature, people report, “*a feeling of coming closer to themselves, to their problems and existential meanings*” [4], “*more space to think about things and to think about them differently*” [5], and “*some kind of thankfulness, thankfulness for all the beauty and for being able to experience this*” [6]. These observations, along with results of more controlled studies [7,8], suggest that exposure to nature might influence coping with psychological distress.

The present paper investigates whether exposure to nature might influence coping with psychological distress. Scientifically, addressing this question is relevant from a fundamental perspective because coping may be one of the psychological mechanisms underlying the restorative effects of nature. With regard to societal application, it is also important to investigate whether nature can influence coping with psychological distress, because it can inform, for example, nature therapy practice [9]. In what follows we will first provide a short overview of how coping is operationalized in this paper. Next we discuss whether nature could influence cognitive coping with psychological distress.

### 1.1. Coping

Coping can be defined as the behavioral and cognitive strategies that are employed to manage psychological distress [10,11]. In the present paper, we will specifically focus on cognitive strategies, and will consequently use the term “cognitive coping” in the following text. Cognitive coping is important as it helps people to deal with, work through, and process difficult thoughts and emotions which in the end helps to maintain wellbeing and mental health. People can apply different cognitive coping strategies [12]. Cognitive coping strategies can be more emotion-focused (i.e., focused on regulating negative emotional reactions to a stressor) or more active-focused, which is also called problem-focused (i.e., focused on eliminating the causes of stress) [12,13]. An example of cognitive emotion-focused coping is for instance trying not to think about a problem. An example of cognitive problem-focused coping is for instance trying to think of possible solutions to a problem.

Some previous research implies that active-focused coping strategies are related to a better mental condition [14], compared to the emotion-focused coping styles [15]. However, other studies posit that it cannot be claimed that one coping strategy is better than the other, but instead the application of different strategies is important [16,17,18]. That is, when a cognitive coping strategy is used rigidly and people are thus not able to apply the strategy that is most helpful in a given situation, this creates a vulnerability for psychopathology [16,17,19]. An important distinction here lies between coping styles as a dispositional trait and coping styles in a specific context or state, a.k.a. there is a distinct difference between ‘trait’ and ‘state’ coping. In a given situation or ‘state’, not one coping style is maladaptive per se [16,17,18]. However, when that coping style is used rigidly and thus becomes a ‘trait’, it can become maladaptive. For instance; a problem-focused strategy like “thinking that there are worse things that can happen” can be adaptive in a moment when you are caught in the rain without an umbrella. However, when it becomes a “trait” coping style it can represent a trivializing of your own feelings. Additionally, vice versa, it can be helpful to apply an emotion-focused coping style like “trying to think of something else” when a loved one is in surgery and there is nothing you can do. However, when it becomes a “trait” coping style it represents not being able to face your feelings at all. People who only apply one strategy for any given situation are thus more vulnerable for mental disorders [16,17,19].

For the present research we were interested in whether cognitive coping with psychological distress is context dependent. More specifically, we were interested in whether the natural environment can influence how people cope with psychological distress.

### 1.2. Nature and Cognitive Coping

Nature as a restorative environment promotes well-being and may ameliorate mental health problems [1,2,3]. For instance, epidemiological studies have shown that people report feeling healthier when more nature is present in their neighborhoods [20]. Likewise, experimental studies have shown that walking in nature or merely viewing (images of) nature can expedite recovery from stress and mental fatigue [3,21]. However, how these restorative benefits might influence cognitive coping with psychological distress remains to be studied.

A growing number of studies have documented how exposure to nature has cognitive benefits like replenishment of attention [3]. These cognitive benefits are most often explained by Attention Restoration Theory (ART), which was developed by Kaplan and Kaplan [22]. They state that natural elements engage attention in an effortless manner (soft fascination), allowing the mind to rest its directed attention system [22]. We use our directed attention system for instance, to prioritize tasks in daily life and solve problems. Imaginable, this directed attention system can get depleted, resulting in attention fatigue and hence a need for restoration. According to the most empirically supported aspects of ART, exposure to nature replenishes our directed attention capacities by the effortless redirection of attention to natural elements and by a sense of being away from all hassles [3].

Based on ART it seems plausible to expect that exposure to nature can influence cognitive coping with psychological distress, for example because the replenished attentional capacities can be directed towards reflection [23,24]. However, to our knowledge no studies have directly assessed whether exposure to nature can influence cognitive coping with psychological distress. Some previous research has assessed however, how exposure to nature influences psychological distress. Berman and colleagues [25] for instance, studied the effects of a 50-min walk for 20 participants suffering from clinical depression. In this study, they induced psychological distress by asking their participants to think about an unresolved autobiographical memory. Next, participants were randomly assigned to walk in an urban or natural environment. Results showed that participants in the nature condition experienced fewer negative thoughts, and also performed better on a cognitive task, compared to the participants in the built environment. However, participants indicated that they thought about the autobiographical memory to the same extent in both environments.

In a study of Bratman [26] psychological distress was assessed before and after a 90-min walk in either natural or urban surroundings with the rumination subscale of the Reflection Rumination Questionnaire (RRQ), a measure of trait rumination, and by assessing brain activity in the Subgenual Prefrontal Cortex (sgPFC) for 38 participants. They found that both self-reported rumination and brain activity in the sgPFC was less compared to the baseline values on these measures for the participants in the nature (vs. built) group. Finally, Golding, Gatersleben, and Cropley [27] studied how people deal with psychological distress while viewing nature (vs. built) images. First, psychological distress was induced with a presentation task for 58 participants, whereafter they were randomly assigned to either wait patiently without distractions, or to watch a slideshow of natural or urban surroundings. The authors found no significant differences between conditions on how much participants had thought about the presentation task.

These studies provide information about the influence of exposure to nature on the quantity of thoughts, whether or not related to psychological distress. Building upon this work, the question arises how participants have coped with their psychological distress during the nature (vs. built) exposure. It is for instance unknown if coping strategies were used and if so, to what extent they were used and how many strategies were combined. Therefore, the present research was developed to build further upon previous research to gain a more in depth understanding of whether exposure to nature can influence Quantity (how much do people cope?), Content (To what extent are specific coping strategies displayed?). and Quality (How many different strategies are combined?) of ‘state’ cognitive coping with psychological distress.

With the novelty of the present research in mind, we wanted to conduct the research in a controlled lab setting. The outcome measure of interest was ‘state’ cognitive coping. However, to the researchers’ knowledge, the only available instruments to assess cognitive coping were trait measures. The authors of this paper aimed to address this limitation by creating a State Coping Scale (SCS), based on the most extensively used trait coping measure in the Netherlands, the well-validated Utrecht Coping List (UCL) [18,28].

### 1.3. The Present Research

To investigate whether nature exposure influences’ the Quantity, Content, and Quality of ’state’ cognitive coping with psychological distress, three controlled lab studies were conducted that studied how people cognitively cope with an autobiographical memory of serious regret while viewing a video of either natural or built surroundings. The research samples in the present paper are from the “healthy” university student population. However, it is commonly known that psychological distress is highly prevalent amongst university students, as is for instance indicated by high prevalence rates of symptoms of depression [29]. It is thus important to study factors that can influence how university students cope with psychological distress, to inform interventions that aim to ameliorate their mental health.

## 2. Study 1

### 2.1. Materials and Methods

#### 2.1.1. Participants

A priori, a power analysis was conducted using G*Power (3.1) for an ANOVA with fixed effects (main effects and interactions), a medium effect size (f = 0.30), and 80% power [30]. The suggested sample size was 190 participants or more. All participants (*n* = 198) were University students, mostly in the Social Sciences. Their age ranged from 18 to 30 years old (M = 21.11, SD = 2.33), and 71.00% were female (*n* = 141). Depressive symptoms scores ranged from 0 to 47 (M = 13.12, SD = 9.59), and 32.80% (*n* = 65) of the participants reported clinically significant depressive symptoms (CES-D score ≥ 16). Participants were recruited through direct recruitment on campus and via the SONA system, a cloud based subject pool software for Universities. Students received EUR 7.50 or 46 research credits as compensation for their participation. The eligibility requirements for this study were (a) being a student, (b) being 18 years or older, and (c) fluent in Dutch, moreover people were instructed not to participate if they (i) had a diagnosis of a psychiatric disorder in the past year, and (ii) received psychotherapy or used antidepressants in the past year. Conform approval by the Institutional Research Ethics Committee, an informed consent form was completed prior to the start of the study and a debriefing was provided after the study. Before analysis, *n* = 7 participants were excluded because they responded with ‘no’ to the question whether they had paid attention to the environmental video. This resulted in a final sample of *n* = 191, of which *n* = 97 participants were in the built condition and *n* = 94 in the nature condition.

#### 2.1.2. Measures

*Randomization Check.* A measure of trait psychological distress, i.e., depressive symptoms was included as a randomization check. To be able to assess whether the assignment of groups was indeed random, and thus that if effects were found they could probably not be explained by one group already suffering less or more from psychological distress at the start of the study.

To assess whether the participants in both groups differed on the extent to which they suffered from depressive symptoms, the Center for Epidemiological Studies-Depression Scale (CES-D) [31] was administered at the start of the questionnaire. This widely used scale, which consists of 20 items, is considered an accurate and valid measure of depression in the general population [32] (p. 128). Participants are asked to indicate how often they have felt or behaved certain ways during the past week on a 4-point scale. Examples of the items are: ‘I was bothered by things that usually don’t bother me’, and ‘I did not feel like eating; my appetite was poor’. The reliability of the CES-D is good with a Cronbach’s alpha of 0.89 in the original study [32]. The Cronbach’s alpha for the present study was 0.93.

*Cognitive Coping.* A questionnaire to assess state cognitive coping was developed based on the Utrecht Coping List (UCL), a well-validated Dutch measure for trait coping styles [18,33]. The UCL is frequently used for diagnostic purposes of trait coping strategies in clinical practice in the Netherlands [28]. The original UCL has seven subscales of which the four cognitive subscales were selected for this study: active problem solving, avoiding, passive reacting, and reassuring thoughts. Active problem solving and reassuring thoughts both represent problem-focused coping styles [13,34]. Avoiding and passive reacting both represent emotion-focused coping styles [16]. For an overview of the specific items please see Table 1. The UCL items were modified to assess to what extent participants coped with the autobiographical recall memory while watching the environmental video. We have named the result of this modification, the State Coping Scale (SCS). Examples of modified items were “I have tried to see the humoristic side of it” and “I have tried to think of it as little as possible”. Similar to the UCL, answer categories for the SCS ranged from not at all to very much on a 5-point scale.

*Quantity*. Conforming to the guidelines of the original UCL, a continuous variable was computed as a measure of total cognitive coping Quantity (min. 12, max. 60) by adding up the individual item scores.

*Content*. Due to the innovative nature of modifying items from a “trait” to a “state” manner, a factor analysis was used to check if the subscales matched the original four subscales of the UCL. The factor analysis with varimax rotation revealed three subscales instead of the four of the UCL. More specifically, the items of the subscales “active problem solving” and “reassuring thoughts” resulted in one factor. Both subscales represent active- focused or problem-focused coping styles [13,34], therefore we named this factor “Active” coping in the present paper. We named the other two emotion-focused factors “Passive” coping and “Avoidance” coping [13,16]. The measure of cognitive coping content thus comprised of three subscales: “Active”, “Passive”, and “Avoidance” coping. The higher the score on a subscale, the more the respective coping strategy has been used. Reliability of the subscales “Active” coping and “Passive” coping was sufficient with a Cronbach’s alpha for “Active” coping of 0.84 and for “Passive” coping of 0.70. For “Avoidance” coping the reliability was moderate with a Cronbach’s alpha of 0.60. Please see Table 1 for an overview of the factor analysis and its overlap and differences compared to the original UCL subscales.

*Quality*. First, the subscales (“Active”, “Passive”, and “Avoidance”) were dichotomized into 0 (did not use this specific strategy) and 1 (strategy used), and the sum of the score was calculated on the three subscales. This resulted in a variable with 4 levels: no strategy used, one strategy used, two strategies used, and three strategies used.

This study was conducted as part of educational projects for psychology students, and therefore included additional measures to accommodate teaching requirements. This paper will not report on any of these additional measures, because this goes beyond the scope of the present paper.

#### 2.1.3. Experimental Manipulations

*Psychological distress induction.* Conform a procedure used by others [25,35,36] an autobiographical recall task was used to induce psychological distress which participants would need to cope with. More specifically, participants were asked the following four questions: “Think of a situation in which you said or did something that you seriously regret”. (1) Recall the situation, what was it like?”; (2) “Think of the emotions you had in that situation. What emotions were those?”; (3) “Why did you have those emotions?” and (4) “What might be underlying reasons for these emotions?”. Participants were given 60 s to think about each question and type a couple of words to describe their answers before the next question followed automatically.

*Environmental videos.* The environmental videos showed a moderately paced walk through a natural and built environment, from the first-person perspective. The built walk was filmed in a calm side-road turning into a pedestrian street in the city of The Hague, the Netherlands, in summer. There were some natural elements such as trees and flowerbeds, but mostly built elements such as streets, residential houses, and vehicles. The route was chosen to clearly represent an attractive built environment while not being overloaded with stimuli. The soundscape is easily identifiable as a city without being exceptionally noisy.

The nature video depicted a walk through a summer forest of the “Nationaal Park de Hoge Veluwe”. It is a slightly curvy, sandy, paved forest trail passing leafy trees and greenery. The soundscape is that of twittering birds and the faint sound of footsteps of the filmmaker. The soundscape of the nature video was amplified to approximately match the average loudness of the built video. A screenshot of the videos is depicted in Figure 1.

#### 2.1.4. Procedure

When arriving at the lab, participants were asked to read the information letter, ask any questions they might have, sign the informed consent, and provide the researchers with information regarding payment for either credits or money. Hereafter, participants were asked to store their phone in a safety box, so they would not be tempted to look at their phones during the experiment. The questionnaire was administered in private cubicles using Qualtrics, a web-based survey tool for data collection. First, participants completed the CES-D. Subsequently, psychological distress was induced using the autobiographical recall task. After this induction participants were randomly assigned to either the nature or the built video. Both environmental videos comprised of a 4-min walk. After the environmental video, the State Cognitive Scale (SCS) items were administered to the participants to assess how “they dealt with their thoughts about their regretful memory while watching the video”. Finally, at the end of the questionnaire, participants were asked some demographic questions such as age and sex.

After completion of the questionnaire, participants were debriefed and provided with information about who to contact in case they had any negative feelings resulting from the study. They were also instructed not to share any details about the study with their fellow students. They were handed back their phones and payment (46 credits or EUR 7.50) was processed. This study had a total duration of approximately 45 min.

#### 2.1.5. Statistical Procedure

We used univariate ANOVA’s to test our theoretical predictions. The assumption of homogeneity of variances for this analysis was met for all measures. Kolmogorov–Smirnov tests of normality indicated that the assumption of normality was violated. Therefore, analyses were performed utilizing bootstrapping with *n* = 1000 random samples [37]. Non-parametric tests were used to analyze the randomization and manipulation checks. All tests were administered in IBM SPSS statistics (v26) using an alpha-level of 0.05. One outlier (>3 SD) was found, for the “Passive” coping strategy. Analysis with and without this outlier did not result in differences regarding the significance of the effects. We therefore retained this outlier in the analyses.

### 2.2. Results

#### 2.2.1. Randomization Checks

To assess if the randomization over the two conditions resulted in comparable groups, non-parametric independent sample tests were computed for the following variables: age, sex, and depressive symptoms. Participants in the nature group scored higher on depressive symptoms compared to the participants in the built groups. Consequently, the CES-D depressive symptoms score was included as a covariate in the outcome analyses. No significant differences were found for the other variables. Please see Table 2 in for the descriptive statistics and test statistics of these randomization checks.

#### 2.2.2. Cognitive Coping

To test our predictions, bootstrapped univariate ANCOVA’s were computed with the respective outcome measure as the dependent variable, condition as the independent variable and CES-D score as a covariate.

*Quantity.* Participants in the nature condition (M = 22.31, SD = 7.12) had a higher “Total Coping” score compared to the participants in the built condition (M = 17.92, SD = 6.46): F (1, 188) = 15.89, *ρ* < 0.001, ɲ_p_^2^ = 0.08.

*Content.* Participants in the nature group (M = 10.99, SD = 4.58) used more “Active Coping” compared to the participants in the built group (M = 8.43, SD = 3.80): F (1, 188) = 15.58, *ρ* < 0.001, ɲ_p_^2^ = 0.08. Participants in the nature group (M = 4.84, SD = 2.27) also used more “Avoidance Coping” compared to the participants in the built group (M = 3.85, SD = 2.09): F (1, 188) = 7.52, *ρ* = 0.01, ɲ_p_^2^ = 0.04. No significant difference between participants in the nature group (M = 6.48, SD = 2.53) and built group (M = 5.64, SD = 2.55) was found for “Passive Coping”: F (1, 188) = 2.66, *ρ* = 0.10, ɲ_p_^2^ = 0.01. Please see Figure 2 for a graphical depiction of the effects of the continuous variables.

*Quality.* The analysis with the categorical variable “Number of Strategies” as outcome measure revealed as well that participants in the nature group (M = 2.33, SD = 1.00) used more strategies (min 0, max 3) compared to the participants in the built group (M = 1.54, SD = 1.17): F (1, 188) = 20.52, *ρ* < 0.001, ɲ_p_^2^ = 0.10.

Please see Figure 2 for a graphical depiction of the effects of the State Coping Scale (SCS) variables. Please see Table A2 for a detailed overview of the descriptive and test statistics of the outcome measures.

### 2.3. Conclusion

Results showed a higher cognitive coping Quantity and a higher cognitive coping Quality for the nature (vs. built) condition. Regarding cognitive coping Content, the nature (vs. built) condition elicited more “Active” and more “Avoidance” coping.

## 3. Study 2

Study 2 was conducted as a pilot study with additional measures to assess the extent to which participants had thought about their memory of serious regret and the vividness of this memory while viewing the environmental videos. More information about the extent was included to foster better interpretation of our findings in relation to previous research. Additional items about vividness of the memory of serious regret were included, because vividness can be experienced as unpleasant. On the other hand, accessibility of vivid cognitions and emotions is important for psychotherapy, where it is an essential requirement to orchestrate successful treatment interventions for numerous evidence based treatments [38,39,40]. Finally, to be able to assess whether the nature and built video were soundly chosen to represent ecologically valid surroundings, four items were included in Study 2 as a manipulation check of the environmental videos.

### 3.1. Method

#### 3.1.1. Participants

All participants (*n* = 59) in this pilot study were University students, mostly in the Social Sciences. Their age ranged from 18 to 58 years old (M = 22.39, SD = 5.12), and 74.60% were female (*n* = 44). Depressive symptoms scores ranged from 1 to 37 (M = 12.97, SD = 8.92), and 35.60% (*n* = 21) of the participants reported clinically significant depressive symptoms (CES-D score ≥ 16). Participants were recruited in a similar manner as for Study 1 and the eligibility requirements were the same as well. As compensation participants received €10.00 or 60 research credits. Before analysis no participants were excluded because no one had responded with ‘no’ to the question whether they had paid attention to the environmental video.

#### 3.1.2. Measures

*Randomization Check.* Similar to Study 1, depressive symptoms (CES-D, Cronbach’s alpha: 0.92) were selected and assessed to test whether random assignment to the nature and built condition was successful on a trait measure of psychological distress.

*Environmental Videos Check.* As an addition to Study 1, Study 2 included a check for the environmental videos. Participants were asked to rate how relaxing, boring, tiresome, and beautiful, they had perceived their video, using a visual analogue scale from 0 (not at all) to 100 (completely).

*Cognitive Coping.* Cognitive coping was again assessed with the SCS, in a similar manner as Study 1.

*Quantity.* As a measure of cognitive coping quantity, the “Total Coping” score was again computed.

*Content.* As a measure of cognitive coping Content, again the subscales of “Active”, “Passive”, and “Avoidance” coping were computed. Reliability of the subscales “Active” coping and “Avoidance” coping was sufficient with a Cronbach’s alpha for “Active” coping of 0.70 and 0.63 for “Avoidance” coping. For “Passive” coping reliability was insufficient a Cronbach’s alpha of 0.42.

*Quality.* To assess cognitive coping quality the “Number of Strategies” variable was again computed.

*Extent.* As an addition to Study 1, Study 2 included an extra question related to the extent to which participants had thought about their memory of serious regret during the video: ‘During the video, to what extent did you think of the situation that you regret?’. This question was answered on a 10-point scale from 0 (not at all) to 10 (a lot/very).

*Vividness.* Study 2 also included three questions about vividness of the psychological distress experience: (1) ‘How vivid was your recall of the situation?’, (2) ‘How vividly did you feel the emotions you had experienced?’, and (3) ‘When you thought of the recalled memory, how pleasant were those thoughts?’. These questions were based on the items from [41], whose items were adapted from the Alcohol Craving Experience questionnaire [42,43]. Answers were given on a 10-point scale from 0 (not at all) to 10 (a lot/very).

Similar to Study 1, this study was conducted as part of educational projects for psychology students. It therefore included additional self-report measures and a measure of brain activity (functional Near-InfraRed Spectroscopy: fNIRS) to accommodate teaching requirements. This paper will not report on any of these additional measures, because this goes beyond the scope of the present paper.

#### 3.1.3. Experimental Manipulations

Both the psychological distress induction and the environmental videos for Study 2 were the same as those of Study 1.

#### 3.1.4. Procedure

The procedure for Study 2 was similar to that of Study 1, with the addition of the following: at the end of the questionnaire participants were asked about three questions to provide a more general notion of the manner in which participants had processed their personal memory during the video and how vivid their experience was. Finally, they were asked to rate how relaxing, boring, tiring and beautiful the videos were. This study had a total duration of approximately 60 min.

#### 3.1.5. Statistical Procedure

For all measures, the assumption of homogeneity for the univariate ANOVA’s was met. Kolmogorov-Smirnov tests of normality indicated that the assumption of normality was violated. Therefore, analyses were performed utilizing bootstrapping with *n* = 1000 random samples [37]. Non-parametric tests were used to analyze the randomization and manipulation checks. All tests were administered in IBM SPSS statistics (v26) and an alpha-level of 0.10 was used in this study due to the small sample size of this pilot study.

### 3.2. Results

#### 3.2.1. Randomization and Environmental Videos Checks

To assess if the randomization over the two conditions resulted in comparable groups, non-parametric independent sample tests were computed for the following variables: age, sex, and depressive symptoms. No significant differences were found. Please see Table 2 for the descriptive statistics and test statistics of these randomization checks.

Ratings of the environmental videos were analyzed as a manipulation check to see whether they differed on aspects inherent to natural surroundings (relaxing, beautiful) and were similar on unintended aspects (boring, tiresome). In line with expectations, non-parametric independent samples tests revealed that the nature video was perceived as more relaxing (Mann–Whitney U (59) = 575.00, *ρ* = 0.03, Cohen’s d = 0.58) and beautiful (Mann–Whitney U (59) = 630.50, *ρ* = 0.003, Cohen’s d = 0.84) compared to the built video. Moreover, as expected no significant differences were found regarding how tiresome (Mann–Whitney U (59) = 437.00, *ρ* = 0.98, Cohen’s d = 0.01) and boring (Mann–Whitney U (59) = 498.00, *ρ* = 0.33, Cohen’s d = 0.25) the videos were rated. These results indicate that the video manipulation was successful and therefore support the choice for these videos as stimuli material. Please see Table A1 for an overview of the descriptive and test statistics of these environmental video checks.

#### 3.2.2. Cognitive Coping

*Quantity.* No significant difference was found between the nature group (M = 19.63, SD = 5.52) and the built group (M = 17.52, SD = 6.05) for “Total Coping”: F (1, 57) = 1.97, *ρ* = 0.17, ɲ_p_^2^ = 0.03.

*Content.* Participants in the nature group (M = 4.93, SD = 2.18) used more “Avoidance” coping compared to the participants in the built group (M = 3.86, SD = 2.25): F (1, 57) = 3.45, *ρ* = 0.07, ɲ_p_^2^ = 0.06. No significant difference was found between the nature group (M = 9.20, SD = 3.23) and the built group (M = 8.59, SD = 3.24) for “Active” coping: F (1, 57) = 0.53, *ρ* = 0.47, ɲ_p_^2^ = 0.01. Additionally, no significant difference between participants in the nature group (M = 5.50, SD = 1.63) and built group (M = 5.07, SD = 1.58) was found for “Passive” coping: F (1, 57) = 1.06, *ρ* = 0.31, ɲ_p_^2^ = 0.02.

*Quality.* The analysis with the categorical variable “Number of Strategies” revealed a significant difference, where participants in the nature condition (M = 2.10, SD = 1.12) used more strategies (min 0, max 3) compared to the participants in the built condition (M = 1.59, SD = 1.24): F (1, 57) = 2.78, *ρ* = 0.10, ɲ_p_^2^ = 0.05.

Please see Figure 3 for a graphical depiction of State Coping Scale (SCS) results.

*Extent.* Non-parametric independent samples tests revealed that participants in the nature condition thought about their memory of serious regret to a greater extent compared to the participants in the built condition: Mann–Whitney U (59) = 542.50, *ρ* = 0.08, Cohen’s d = 0.43.

*Vividness.* Non-parametric independent samples tests revealed that participants in the nature condition experienced the situation more vividly: Mann–Whitney U (59) = 586.50, *ρ* = 0.02, Cohen’s d = 0.63, and that they experienced the accompanying emotions more vividly: Mann–Whitney U (59) = 551.00, *ρ* = 0.07, Cohen’s d = 0.47. No significant difference was found for how pleasant their thoughts about the recalled memory were: Mann–Whitney U (59) = 346.00, *ρ* = 0.16, Cohen’s d = 0.36.

Please see Table A2 and Table A3 for a detailed overview of the descriptive and test statistics of all outcome measures.

### 3.3. Conclusion

Results showed no difference between the groups regarding cognitive coping Quantity. However results did reveal that participants had thought about their memory of serious regret to a greater extent in the nature (vs. built) condition. Results also revealed a higher cognitive coping Quality for the nature (vs. built) condition. Regarding cognitive coping Content, the nature (vs. built) condition elicited more “Avoiding” coping. No differences were found for “Active” and “Passive” coping. Finally, results showed that the memory of serious regret was experienced more vividly in the nature (vs. built) condition, but participants did not experience this as more unpleasant.

## 4. Study 3

In Study 3 we assessed similar measures as in Study 2 but with a larger sample size to accommodate the limitation of low statistical power of the pilot study. Additionally we aimed to see whether the results of Study 1 and/or Study 2 regarding a higher cognitive coping Quantity and Quality, and a higher extent and vividness could be replicated. We therefore preregistered the study before the start of data collection. The study was preregistered at the Open Science Framework (OSF) at 3 April 2019, at 12:07 p.m. Regarding the State Coping Scale (SCS), it was hypothesized in this preregistration that participants would use more coping, and specifically more adaptive and more maladaptive coping in the nature condition, compared to the built condition. It was also expected that participants in the nature (vs. built) condition would rate their experience as more pleasant. However, new insights in studying the “state” vs. “trait” coping literature resulted in re-evaluation of the suitability of the (mal)adaptive terminology, which was consequently aborted. Moreover, with the novel nature of the SCS in mind, we came to the conclusion that a factor analysis was necessary (see Section 2.1.2). Thus, the subscales of “Active”, “Passive”, and “Avoidance” coping were not described in the preregistration.

Study 3 was, similar to Study 1 and Study 2, conducted as part of educational projects for psychology students. It therefore included additional measures to accommodate teaching requirements, which were also preregistered. We will not report on any of these additional measures, because this goes beyond the scope of the present paper. The specific order of the measures and experimental manipulations used in Study 3 is described in Appendix A.

### 4.1. Method

#### 4.1.1. Participants

All participants (*n* = 249) were freshman Social Sciences University students who participated for 46 research credits. Their age ranged from 18 to 30 years old (M = 20.38, SD = 1.99), and 79.00% was female (*n* = 196). Their depressive symptoms scores ranged from 0 to 40 (M = 11.76, SD = 8.08), and 28.51% (*n* = 71) of these students reported clinically significant depressive symptoms (CES-D ≥ 16). Recruitment and eligibility requirements for Study 3 were the same as for Study 1 and Study 2, with the exception that participants were required to speak either Dutch or English to participate. Most participants, namely 68.70% (*n* = 717) completed the Dutch version. Before analysis, two participants were excluded because they responded with ‘no’ to the question whether they were paying attention to the environmental video. This resulted in a final sample of *n* = 247, of which *n* = 124 participants were in the built condition and *n* = 123 in the nature condition.

#### 4.1.2. Measures

*Randomization Check.* Similar to Study 1 and Study 2, depressive symptoms (CES-D, Cronbach’s alpha of 0.89) were selected and assessed to test whether random assignment to the nature and built condition was successful on this measure of trait psychological distress.

*Environmental Videos Check.* In Study 3, participants were asked again to rate how relaxing, boring, tiresome, and beautiful, they had perceived their video, using a visual analogue scale from 0 (not at all) to 100 (completely).

*Cognitive Coping.* Cognitive coping was again assessed with the SCS, in a similar manner as Study 1.

*Quantity.* As a measure of cognitive coping quantity, the “Total Coping” score was again computed.

*Content.* As a measure of cognitive coping Content, again the subscales of “Active”, “Passive”, and “Avoidance” coping were computed. A factor analysis with varimax rotation revealed a division of items in three subscales, similar to the factor analysis of Study 1. Reliability of the subscales “Active” coping and “Passive” coping was sufficient with a Cronbach’s alpha for “Active” coping of 0.78 and 0.76 for “Passive” coping. For “Avoidance” coping the reliability was moderate with a Cronbach’s alpha of 0.67.

*Quality.* To assess cognitive coping quality the “Number of Strategies” variable was again computed.

*Extent.* Similar to Study 2, Study 3 also included the question related to extent: ‘During the video, to what extent did you think of the situation that you regret?’. This question was answered on a ten-point scale from 0 (not at all) to 10 (a lot/very).

*Vividness.* Similar to Study 2, Study 3 also included three questions about vividness of the psychological distress experience during the environmental videos: (1) ‘How vivid was your recall of the situation?’, (2) ‘How vividly did you feel the emotions you had experienced?’, and (3) ‘When you thought of the recalled memory, how pleasant were those thoughts?’. Answers were given on a 10-point scale from 0 (not at all) to 10 (a lot/very).

#### 4.1.3. Experimental Manipulations

Both the psychological distress induction and the environmental videos for Study 3 were the same as those of Study 1 and Study 2.

#### 4.1.4. Procedure

The procedure for Study 3 was similar to that of Study 2. This study had a total duration of approximately 45 min.

#### 4.1.5. Statistical Procedure

For all measures, the assumption of homogeneity for the univariate ANOVA’s was met. Kolmogorov–Smirnov tests of normality indicated that the assumption of normality was violated. Therefore, analyses were performed utilizing bootstrapping with *n* = 1000 random samples [37]. Non-parametric tests were used to analyze the randomization and manipulation checks. All tests were administered in IBM SPSS statistics (v26) using an alpha-level of 0.05.

### 4.2. Results

#### 4.2.1. Randomization and Environmental Videos Checks

To assess if the randomization over the two conditions resulted in comparable groups, non-parametric independent sample tests were computed for the following variables: age, sex, and depressive symptoms. No significant differences were found, with the exception that participants in the nature group scored higher on depressive symptoms compared to the participants in the built groups. Therefore the CES-D depressive symptoms score was included as a covariate in the outcome analyses. Please see Table 2 for the descriptive statistics and test statistics of these randomization checks.

Ratings of the environmental videos were analyzed as a manipulation check to see whether they differed on aspects inherent to natural surroundings (relaxing, beautiful) and were similar on unintended aspects (boring, tiresome). In line with expectations, non-parametric independent samples tests revealed that the nature video was perceived as more relaxing: Mann–Whitney U (247) = 10,227.50, *ρ* ≤ 0.001, Cohen’s d = 0.62; and more beautiful: Mann–Whitney U (247) = 11,968.50, *ρ* ≤ 0.001, Cohen’s d = 1.13, compared to the built video. Moreover, as expected no significant differences were found regarding how tiresome the videos were rated: Mann–Whitney U (247) = 7278.00, *ρ* = 0.530, Cohen’s d = 0.08. However, the built video was rated as more boring compared to the nature video, Mann–Whitney U (247) = 6283.50, *ρ* = 0.016, Cohen’s d = 0.31. These results indicate that the video manipulation was only partly successful. Please see Table A1 for an overview of the descriptive and test statistics of these environmental video checks. 

#### 4.2.2. Cognitive Coping

*Quantity.* Participants in the nature condition (M = 23.59, SD = 7.65) had a higher “Total Coping” score compared to the participants in the built condition (M = 19.86, SD = 6.92): F (1, 244) = 13.49, *ρ* < 0.001, ɲ_p_^2^ = 0.05.

*Content.* Analyses revealed that participants in the nature group (M = 11.72, SD = 4.67) used more “Active” coping compared to the participants in the built group (M = 9.70, SD = 4.11): F (1, 244) = 11.87, *ρ* = 0.001, ɲ_p_^2^ = 0.05. Participants in the nature group (M = 7.24, SD = 3.14) also used more “Passive” coping compared to the participants in the built group (M = 5.76, SD = 2.22): F (1, 244) = 15.25, *ρ* < 0.001, ɲ_p_^2^ = 0.06. No significant difference between participants in the nature group (M = 4.62, SD = 2.16) and built group (M = 4.40, SD = 2.38) was found for “Avoidance” coping: F (1, 244) = 0.16, *ρ* = 0.69, ɲ_p_^2^ = 0.001.

*Quality.* The analysis with the categorical variable “Number of Strategies” as outcome measure revealed that participants in the nature group (M = 2.37, SD = 0.96) used more strategies (min 0, max 3) compared to the participants in the built group (M = 1.91, SD = 1.22): F (1, 244) = 8.85, *ρ* = 0.003, ɲ_p_^2^ = 0.04.

Please see Figure 4 for a graphical depiction of the effects of the State Coping Scale (SCS) variables.

*Extent.* Non-parametric independent samples tests revealed that participants in the nature condition also thought about their memory of serious regret to a greater extent compared to the participants in to the built condition: Mann–Whitney U (247) = 10,666.50, *ρ* < 0.001, Cohen’s d = 0.73.

*Vividness.* Further, the participants in the nature group (compared to the built group) experienced the regretful situation more vividly during the environmental video: Mann–Whitney U (247) = 9800.50, *ρ* ≤ 0.001, Cohen’s d = 0.51, and experienced the accompanying emotions more vividly as well: Mann–Whitney U (247) = 9976.50, *ρ* ≤ 0.001, Cohen’s d = 0.55. No significant difference was found for how pleasant their thoughts about the recalled memory were: Mann–Whitney U (247) = 7591.00, *ρ* = 0.95, Cohen’s d = 0.01.

Please see Table A2 and Table A3 for a detailed overview of the descriptive and test statistics of all outcome measures.

### 4.3. Conclusions

Results showed a higher cognitive coping Quantity and a higher cognitive coping Quality for the nature (vs. built) condition. Regarding cognitive coping Content, the nature (vs. built) condition elicited more “Active” and more “Passive” coping. No significant difference between conditions was found for “Avoidance” coping. Additionally, revealed that participants had thought about their memory of serious regret to a greater extent in the nature (vs. built) condition. Finally, results showed that the memory of serious regret was experienced more vividly in the nature (vs. built) condition, but participants did not experience this as more unpleasant.

## 5. General Discussion

Three lab experiments were conducted to study whether viewing a natural (vs. built) environment could influence cognitive coping with psychological distress. Cognitive coping was assessed on the aspects of Quantity (how much do people cope?), Content (To what extent are specific coping strategies displayed?) and Quality (How many different strategies are combined?). Study 1 and Study 3 revealed that people had a higher total use of coping while watching the nature video, compared to the built video. For Study 2 this effect was not found to be statistically significant. Moreover, results of both Study 2 and 3 showed that participants had thought more about their memory of serious regret during the nature (vs. built) video. Regarding cognitive coping Content, the nature (vs. built) condition elicited more “Active” coping in Study 1 and 3, more “Avoidant” coping in Study 1 and 2, and more “Passive” coping in Study 3. Furthermore, results across all three studies consistently showed a higher coping Quality. That is to say a higher number of different coping strategies was used in the nature video group, compared to the built video group. Additionally, participants also experienced the regretful situation and the corresponding emotions more vividly. However, participants in the nature (vs. built) group did not rate their experience as less pleasant.

There were some inconsistencies in the results however. Concerning cognitive coping quantity all three studies found a higher total use of coping in the nature (vs. built) group was found, but the association was not significant in Study 2. This can probably be explained by the relatively small sample size in Study 2, which resulted in low statistical power. Therefore it is likely that the interpretation of a higher total use of coping is nature is valid. Another inconsistency is that the results regarding the Content, e.g., the specific coping strategies, varied for each study.

The results of the present research imply that viewing nature elicits a higher cognitive coping Quantity and Quality when dealing with psychological distress. It seems that there is no specific cognitive coping strategy that is consistently used more while viewing nature, compared to viewing a built setting. Rather, nature seems to bolster the ability to combine several cognitive coping strategies.

That participants in the nature condition had thought about their memory of serious regret to a greater extent, compared to the participants in the built condition, does not seem to be in line with previous initial studies on the topic of the influence of nature exposure on psychological distress. Refs. [25,26] found fewer negative thoughts and less trait rumination when comparing a walk in nature to a walk in built surroundings, respectively. However, the result of [25] was a general assessment of thoughts, for thoughts related to the recalled memory they found no difference between the groups. Moreover, the study of [26] did not induce psychological distress. Their results were found on a post walk compared to baseline trait rumination scale and in a very specific brain region. The study of [27] found no significant differences between conditions (nature, urban, no distractions) on how much participants had thought about the psychological stress induction.

The present research was, to our knowledge, the first to assess the influence of viewing nature on cognitive coping with psychological distress.

### 5.1. Proposed Framework

In the present research, participants reported to have thought about their memory of serious regret more while viewing the nature (vs. built) video and reported that this experience was more vivid. This does not sound particularly pleasant, yet somehow participants did not rate their experience as less pleasant. Additionally, both Coping Quantity and Quality were higher while viewing the nature (vs. built) video. This might sound straining to cope more and to combine more specific strategies, yet however participants rated their experience of viewing nature as more relaxing compared to viewing built surroundings. How can we understand all of this? Our results suggest a two-step pathway of how the restorative effects of viewing nature relate to the cognitive coping results found in the present research. As a first step, based on Attention Restoration Theory (ART) it follows that contact with nature directly reduces psychological distress, because your mind gets distracted by the natural elements that engage attention effortlessly [3,22]. This first “peaceful mind” step thus entails that in nature we at first think less about whatever we needed to cope with, until our attentional capacities are somewhat replenished.

The second step follows after the directed attention system is somewhat replenished. It entails that with the renewed capacity for directed attention the mind can wander to internal processes (e.g., feelings and thoughts about the psychological distress) and hence provides the optimal setting for reflection [23,24]. This “peaceful soul” step thus posits that contact with nature creates space to think and that people consequently think more about whatever they need to cope with, cope more and combine more cognitive coping strategies. Thinking more about the experienced psychological distress might not be directly pleasant. However, it might ultimately be relieving and relaxing, because it allows people to truly process whatever is occupying their minds [24,44,45].

### 5.2. Limitations

Inevitably this study design had limitations. First, the present study only assessed cognitive measures while the experience of psychological distress of course also concerns affect. Cognitive coping is a cognitive process, but for future research, it would be recommended to include measures of affect when studying cognitive coping with psychological distress.

Second, the environmental videos had a relatively short duration of 4 min to study the proposed two-step “peaceful soul” pathway. From previous research it is known that the restorative effects of nature exposure occur quickly, with exposure duration of less than 5 min having the largest effect sizes [46]. However, we do recommend to study longer exposure durations in order to see if the effects hold up. Additionally, the present study only included one video of each environment. These videos were chosen for their ecological validity, but we cannot be sure that the present results can be generalized to “nature” and “built” settings in general. We would recommend to use multiple operationalizations of these settings to foster generalization.

Third, we created our own State Coping Scale (SCS), which is not yet a validated measure (even though it was based on the well validated UCL) and might therefor not be sufficiently reliable. However, reliability of the three factors of the SCS were moderate to high in our studies with a sufficient sample size (Study 1 and Study 3). Moreover, the scores on the three specific coping strategies and the total use of coping scores are consistent in the three studies. In addition, for all SCS related variables, the means were consistently higher in the nature group, compared to the built group. Finally, the factor analysis of the studies with sufficient sample sizes for this purpose, Study 1 and Study 3, revealed the exact same three factors, with the exact same specific items. Taken together, we find that our results can be relied upon in good reason. Of course, it is strongly recommended to further develop the SCS in future research to improve reliability and assess its validity with measures of both concurrent and convergent validity.

Finally, the present research used digital nature instead of actual nature. A meta-analysis and systematic review observed that, even though the effect size is larger for studies that expose participants to real nature, lab studies using nature imagery show robust restorative benefits as well [2,3]. We thus suspect that the presented results might be an underestimation of the “real-life” effects. Future research that directly compares the influence of exposure type is recommended. Future research about different degrees of naturalness is also warranted, both for studies that use digital nature and studies that use actual nature. The present research used a built video that contained natural elements such as trees and flowerbeds; and the nature video was a national park that was well-kept. For future research it would be interesting to further investigate degree of naturalness as a factor of influence in restoration.

### 5.3. Implications and Future Perspectives

Based on the increasing knowledge of the beneficial health effects of nature [1], initiatives that integrate nature in clinical settings are booming [47]. For instance, nature is implemented in the waiting room of healthcare facilities [48,49], in the therapist’s office [50], and even in nature-based therapy modules [51,52]. The present research can inform clinical practice about how viewing nature seems to make distressing thoughts and feelings more accessible when someone is psychologically distressed.

This implies that there might be some circumstances that it would not be recommended to seek contact with nature, as for instance when someone is already overwhelmed by their emotions. On the other hand, it is especially relevant for psychotherapy, where accessibility of distressing cognitions and emotions is an essential requirement to orchestrate successful treatment interventions for numerous evidence based treatments [38,39,40]. However, the presence of a therapist is then extra important when considering psychotherapy in this “Walk and Talk” format to provide support and prevent someone from being completely caught up in their psychological distress. For future research it would be interesting to learn more about the “inner” experiences of people during exposure to nature, and specifically clients, when studying the value of nature exposure for clinical practice. It might be that integrating nature in psychotherapy can enhance therapy effectiveness because it bolsters reflection. It might also be plausible that therapy effectiveness can be enhanced, because certain personal characteristics bolster or constrain the effectiveness of nature interventions in clinical practice. For example, it seems that higher levels of depressive symptoms enhance the restorative benefits of nature [53]. Additionally, a higher nature connectedness seems to serve as a stronger predictor for mental health than the duration of nature exposure [54]. There are also indications that nature connectedness can be increased for people with a low connection to nature, which might benefit clinical interventions [55]. Future research is needed to further our understanding of such personal characteristics and processes; and how these influence psychotherapy effectiveness.

Regarding implications for environmental psychology theory, Attention Restoration Theory (ART) is not the only important theoretical framework in restorative environments research. Stress Reduction Theory (SRT) also has a substantial evidence base regarding the affective benefits of nature exposure [56]. It would be interesting to see whether a stress induction study would reveal similar influences of nature exposure on cognitive coping. It is then important to distinguish the mood state stress from psychological distress. A mood state of stress can be a general feeling of unrest or tension, whereas psychological distress entails struggles with oneself. It could be possible that the mood state someone is in determines whether someone experiences just the “peaceful mind” or also “the peaceful soul”. That a persons’ characteristics or mood state can influence the interaction with the environment is considered in the transactional perspective of human environment interactions [57]. This transactional perspective goes beyond the traditional ART and SRT frameworks and might be especially relevant for the study of restorative environments research in clinical settings and/or populations.

Finally, a natural environment is not the only restorative environment [58]. Settings such as libraries, picturesque villages or even certain rooms at home can be restorative to some degree. It might be that the effects found in this paper could also be replicated in settings that are not made up off natural elements. However, it does seem that the natural environment “does” something to us humans that entails more than just restoration [59]. It would be interesting to see what specific aspects of nature drive the “peaceful mind” and/or “peaceful soul” steps. Addressing this question in more ecologically representative settings, for example “Walk and Talk” therapy, would help to disentangle the mechanisms that drive the restorative effects of nature.

## 6. Conclusions

Despite the fact that there is still much left to discover, the present research demonstrates that the environment we are in influences how we cope with our personal struggles. It demonstrates that viewing nature lets your mind run free.

## Figures and Tables

**Figure 1 ijerph-18-08842-f001:**
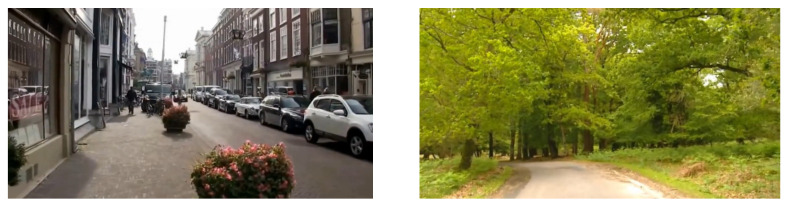
Screenshots of the two environmental conditions, built (**left**) and nature (**right**).

**Figure 2 ijerph-18-08842-f002:**
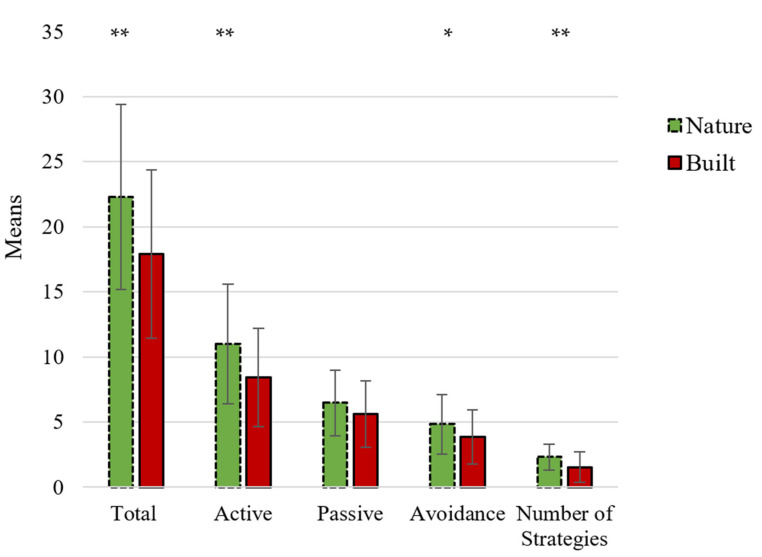
State Coping Scale (SCS) scores for the nature group and built group for Study 2. * *p* < 0.05 and ** *p* < 0.001.

**Figure 3 ijerph-18-08842-f003:**
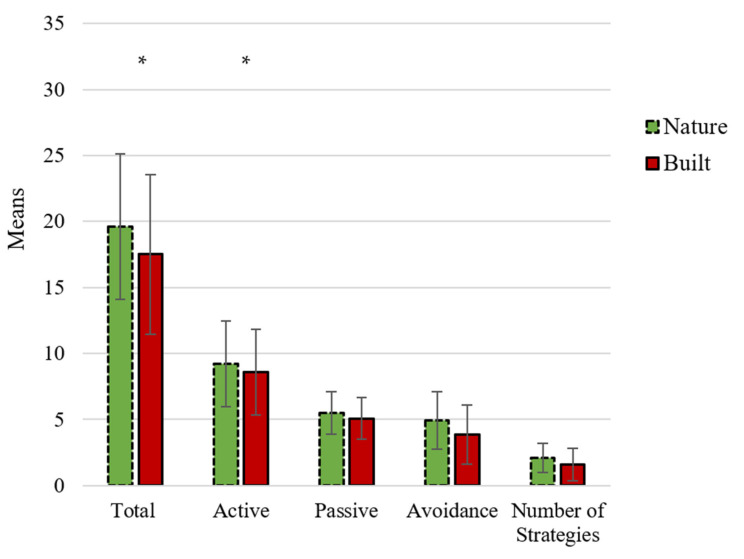
State Coping Scale (SCS) scores for the nature group and built group for Study 2. * *p* < 0.10.

**Figure 4 ijerph-18-08842-f004:**
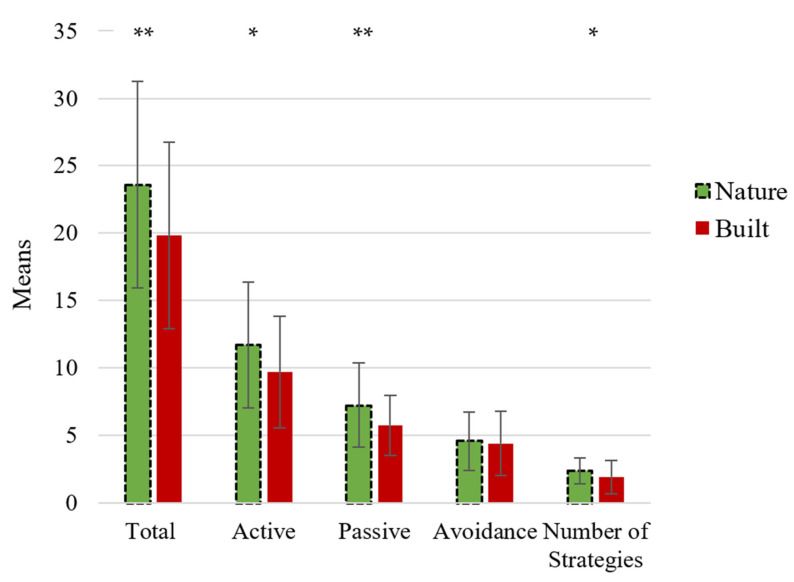
State Coping Scale (SCS) scores for the nature group and built group for Study 3. * *p* < 0.05 and ** *p* < 0.001.

**Table 1 ijerph-18-08842-t001:** Overview of the State Coping Scale (SCS) with the subscales for the present study and the original subscales of the Utrecht Coping List.

Factor Analysis Study 1 and Study 3	Subscale Original UCL	Items Cognitive Coping Scale
Avoidance coping	Avoidance	Tried to think of it as little as possible
Passive coping	Passive expectancy	Brooded over it
Active coping	Active problem solving	Tried to see the humoristic side of it
Passive coping	Passive expectancy	Let myself completely be controlled by it
Active coping	Passive reaction pattern	Realised that other experience difficulties as well sometimes
Avoidance coping	Avoidance	Tried to think of something else
Active coping	Passive reaction pattern	Encouraged myself
Active coping	Active problem solving	Thought of different possibilities to solve the problem
Active coping	Passive reaction pattern	Realised that there are worse things that can happen
Passive coping	Passive expectancy	Felt gloomy about it
Passive coping	Avoidance	Given into it
Active coping	Active problem solving	Thought of problems as challenges

**Table 2 ijerph-18-08842-t002:** Descriptive and Test Statistics for the Randomization Checks.

		Nature		Built		*U*	*ρ*	Cohens’ d
		Mean	SD	Mean	SD			
Study 1	Sex	69.10% female (*n* = 65)		72.20% female (*n* = 70)		4421.50	0.65	0.05
	Age	21.11	2.21	21.13	2.51	4635.50	0.84	0.03
Depressive Symptoms (CES-D)		14.68	9.69	11.56	9.11	5560.00	0.01	0.39
Study 2	Sex	76.70% female (*n* = 23)		72.40% female (*n* = 21)		453.50	0.71	0.08
	Age	21.60	2.03	23.21	6.97	365.50	0.29	0.28
Depressive Symptoms (CES-D)		14.13	8.88	11.76	8.95	513.00	0.24	0.31
Study 3	Sex	81.30% female (*n* = 100)		75.80% female (*n* = 94)		8045.00	0.29	0.10
	Age	20.25	1.87	20.50	2.12	7242.50	0.48	0.09
Depressive Symptoms (CES-D)		12.78	8.33	10.74	7.79	8786.50	0.04	0.27

## Data Availability

We welcome other researchers to use the data of the present paper for the investigation of additional research questions. We therefore have included a detailed list of all included measures in Appendix A. The corresponding author can be contacted to gain access to the data.

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
