# Peer review of "Viewing Nature Lets Your Mind Run Free: Three Experiments about the Influence of Viewing a Nature Video on Cognitive Coping with Psychological Distress"

_ijerph, 2021, doi:10.3390/ijerph18168842_

Round 1

Reviewer 1 Report

The authors correctly identify the study limitation in the conclusion. There are some outstanding questions related to the determination of coping, but overall, this is a sound study that offers new insights through the inclusion of psychological distress. 

This study proposes a three-part approach to estimating the effect of viewing nature on mood and cognitive coping. The study builds on current research in affect, memory and context studies, but in several ways is an extension of studies in early psychology using ‘introspection.’ Subjects were asked to describe the quality of their experience, recall a difficult experience and report on their mood while viewing different scenes. Measuring cognitive coping in context offers some insight into a more wholistic and broader understanding of the an embodied perception of self rather than the more isolated assessment of ‘trait-based’ coping styles. Outcome measures engaged with total, active, passive and avoidance coping. Results indicate generally higher cognitive coping when subjects viewed nature rather than a built environment. The novel approach here is the consideration of what subjects self-reported while viewing nature. The design of the study does not account for causality, but relies on inference to show the relationship between the duration and vividness of recall of an impactful experience while viewing a natural or built environment. Further study is needed to disambiguate between types of natural environments, whether viewing versus being present in nature offers significantly different advantage to build on the transactional perspective of human environment interaction. 

Reviewer 2 Report

Thank you for submitting a really interesting three-study paper on natural environments and cognitive coping. The paper was well argued and written and helped to expand our understanding. I do have some minor comments on the paper which I hope will help improve the study for publication:

Line 60: The references 12 and 13 are included but it was unclear what alternative term was being referred to here so please include the missing term (it appears to be a typo)

Within the method, an argument is made for why the convenience sample was useful for the study but the rationale for why university students and this being a requirement for participation was needed first in the introduction to set up this argument in the method. Please include some justification for the need to understand coping in university students to help show the purposeful nature of the study rather than it being a convenience sample that was justified post-hoc.  

Line 219: A minor typo as the word ‘therefore’ is missing the letter e.

The Cronbach Alpha for Avoidance in the modified scale was (.60) While it was touched upon in the limitations, more discussion on the low reliability of this subscale and the need for further development of this state measure would be useful in the discussion.

The video for the built environment contains elements of nature which may have been a possible confound. I did appreciate that some checks were made on the traits of the videos but further reflection on the presence of nature in the built video would be useful. Further, the nature video was shot in a clearly heavily managed national park. Further discussion of future research directions looking at differing degrees of natural settings (i.e. built-nature, wild landscapes) and their role as a potential factor in nature and coping would be useful.

Line 400: Nature relatedness is mentioned within the normality checks but the measure was not reported previously anywhere else. I am assuming the measure was removed from the study which is a shame as the role of nature relatedness within coping and the environment would be an interesting one, especially given its mediating role on wellbeing it could perform similarly for coping. Please provide a short rationale/explanation for the removal of the nature relatedness measure as I suspect readers will question this given it is mentioned in the measures used within the study but not reported. Further, something on the removal of the nature relatedness measure for study 3 would be warranted, especially as the study was registered with open science so it would be useful to know what happened here and why it was not reported when the data was gathered (i.e. is it for a separate paper?)

Given the above point and that nature relatedness was included within the design, something on the role of our relationship with nature and cognitive coping in the future research directions would be useful to help cover this aspect given the links to wellbeing this construct has.

Round 2

Reviewer 1 Report

My recommendation was for acceptance. My notes were not to suggest the research is problematic, but that there are some minor points to consider clarifying in the discussion.

Reviewer 2 Report

Thank you for taking the time to address the changes. I feel each was answered meticulously so I will be recommending that the paper now be accepted. 

Author Response

Dear editorial team and reviewers,

We kindly thank the reviewers for taking to time to review the revised manuscript of our paper: Viewing Nature Lets Your Mind Run Free: Three Experiments About the Influence of Viewing a Nature video on Cognitive Coping with Psychological Distress.

Reviewer 2, thank you very much for recommending that the paper now be accepted. 

Best wishes,

the authors